# Comparison Study of Small Extracellular Vesicle Isolation Methods for Profiling Protein Biomarkers in Breast Cancer Liquid Biopsies

**DOI:** 10.3390/ijms242015462

**Published:** 2023-10-23

**Authors:** Yujin Lee, Jie Ni, Valerie C. Wasinger, Peter Graham, Yong Li

**Affiliations:** 1School of Clinical Medicine, St George and Sutherland Clinical Campuses, UNSW Sydney, Kensington, NSW 2052, Australia; yujin.lee2@unsw.edu.au (Y.L.); jie.ni@health.nsw.gov.au (J.N.); peter.graham@health.nsw.gov.au (P.G.); 2Cancer Care Centre, St George Hospital, Kogarah, NSW 2217, Australia; 3Bioanalytical Mass Spectrometry Facility, Mark Wainwright Analytical Centre, UNSW Sydney, Kensington, NSW 2052, Australia; v.wasinger@unsw.edu.au

**Keywords:** breast cancer, extracellular vesicle, biomarker, diagnosis, isolation, liquid biopsy

## Abstract

Small extracellular vesicles (sEVs) are an important intercellular communicator, participating in all stages of cancer metastasis, immunity, and therapeutic resistance. Therefore, protein cargoes within sEVs are considered as a superior source for breast cancer (BC) biomarker discovery. Our study aimed to optimise the approach for sEV isolation and sEV proteomic analysis to identify potential sEV protein biomarkers for BC diagnosis. sEVs derived from BC cell lines, BC patients’ plasma, and non-cancer controls were isolated using ultracentrifugation (UC), a Total Exosome Isolation kit (TEI), and a combined approach named UCT. In BC cell lines, the UC isolates showed a higher sEV purity and marker expression, as well as a higher number of sEV proteins. In BC plasma samples, the UCT isolates showed the highest proportion of sEV-related proteins and the lowest percentage of lipoprotein-related proteins. Our data suggest that the assessment of both the quantity and quality of sEV isolation methods is important in selecting the optimal approach for the specific sEV research purpose, depending on the sample types and downstream analysis.

## 1. Introduction

According to cancer statistics, breast cancer (BC) accounts for 31% of female cancers in the United States in 2023, and it ranks among the top causes of mortality in women, with an estimated 43,170 deaths [1]. Additionally, the incidence rate of BC has been increasing worldwide since 2014, and it is one of the most common cancers among women [2]. One of the primary factors contributing to the death of numerous BC patients is metastatic dissemination within the body. However, current methods for BC detection and prognosis, including tissue biopsy and mammography, are limited by their invasive nature and the static information they provide, making them inadequate for constant disease monitoring and screening. Hence, there is a pressing need to advance non-invasive techniques and identify novel, validated biomarkers for BC in order to effectively decrease the mortality rate linked to this disease [3].

Extracellular vesicles (EVs) are a heterogenous population found in all biofluids, including small EVs (sEVs) ranging from 50 to 150 nm; 100–1000 nm large EVs (lEVs); and the largest vesicle, known as apoptotic bodies, which range from 500 to 5000 nm [4]. Among various populations of EVs, sEVs are abundant in biofluids, have various biologies reflective of their parental cells, and have high stability for a long period of time. sEVs work as an inter-communicator between the cells within tumour microenvironments, transmitting the biological information encapsulated by their vesicles, including RNAs, proteins, and lipids, that can be used as potential cancer biomarkers [5,6,7]. Therefore, sEVs have gained much attention as a promising alternative for non-invasive BC detection and diagnosis methods. Particularly, some proteins within sEVs, such as annexin-A2 [8], programmed death-ligand 1 [9], amphiregulin [10], and insulin-like growth factor-1 [11], are related to angiogenesis, metastasis, and immune escape in BC. Moreover, the proteins found in sEVs encompass approximately 50% of the human proteome and accurately represent the cell types from which they originate [12]. This characteristic makes sEV proteins an optimal candidate for disease-specific studies and the exploration of biomarkers. However, despite some research efforts aimed at identifying sEV protein biomarkers for BC, the lack of standardised sEV isolation methods and the requirement for validation in large cohort studies have hindered the establishment of protein biomarkers for the clinical diagnosis and prognosis of BC.

The ideal standardised isolation method for sEV research should have a simple and fast process; high throughputs, purities, and recovery rates; and low cost. Ultracentrifugation (UC) is the standard method for sEV isolation. However, UC is associated with several drawbacks, including long processing times, low throughputs, and variable purity levels, and it may lead to sEV aggregation, which can affect the reliability and reproducibility of the downstream analysis [13]. Polymer precipitation is another conventional isolation method that can efficiently isolate a high yield of sEVs with a simple process in a short time, but it may result in the production of an increased quantity of contaminants, which can interfere with the subsequent sEV characterisation and further analysis [14]. Currently, no available isolation methods can perfectly isolate sEVs because of their complexity and heterogeneity.

According to a recent study, the combination of two or more isolation techniques may be an effective approach to overcome the limitations of individual methods, depending on the sample types [15]. A hybrid approach, for example, that involves a half-cycle of UC followed by the polyethylene glycol (PEG) isolation method was found to isolate purer sEVs from human serum samples at higher throughputs compared to the isolation method alone [16]. The successful clinical application of this combined approach involves the use of isolated sEVs for aerosol inhalation therapy in the treatment of lung injuries [17]. However, the efficacy of this combination method in isolating sEVs from BC cell lines and plasma samples and its downstream proteomic analysis have not yet been investigated. Furthermore, it is noteworthy that a majority of comparative studies on isolation methods have focused on a single sample type [18,19,20,21,22]. It has been observed that the outcomes of downstream analysis vary considerably depending on the specific isolation method employed for each sample type [19,23,24]. Hence, the selection of an appropriate sEV isolation method becomes crucial, contingent upon the intended downstream analysis and the type of sample under investigation.

In this study, we utilised a combined approach of a half-cycle of UC and Total Exosome Isolation kit (TEI) isolation methods, referred to as UCT. Our hypothesis is that UCT could potentially achieve a greater yield of sEVs with improved purity, resulting in better outcomes in BC protein biomarker discovery through proteomic analysis. We compared three different isolation methods, including UC, TEI, and a combination approach (UCT), to assess each method based on various parameters, including the sEV yield, purity, morphology, size, protein contamination, and proteomic statistical analysis. To the best of our knowledge, this study is the first to compare three isolation methods for determining the optimal technique in sEV proteomic analysis. The evaluation includes a comparison of these methods using both cell-conditioned medium and plasma samples.

## 2. Results

### 2.1. Particle Size Distribution and Concentration of MDA-MB-231 Cell-Line-Derived sEVs by Three Different Isolation Methods

First, to examine the size distribution and particle concentration, NTA was performed on isolates from 60 mL of cell medium supernatants of three different MDA-MB-231 batches obtained through three different methods, including TEI, UC, and UCT. The size distribution of the TEI-derived isolates exhibited the broadest range, from approximately 45 to 535 nm. The UCT isolates had a size range from 55 to 385 nm, while the UC isolates ranged from 45 to 335 nm. Among the three methods, the UCT isolates demonstrated the smallest mean mode size, measuring 140 nm, which was significantly smaller than the mode size of the TEI isolates, at 197 nm (Figure 1A,B). The mean mode size for the UC isolates was 161 nm. In terms of the consistency in the particle size distribution across the three batches, TEI showed variable distributions among the three batches, whereas UC and UCT displayed higher consistency compared to TEI. Additionally, the size distributions of UC and UCT were quite similar. The particle concentration was significantly higher in the TEI-derived isolates compared to UC, but there was no significant difference when compared to UCT (Figure 1C). As described in a previous study [25], the total protein amount (µg/mL) was measured to estimate the non-sEV-related protein (contamination) in the isolates. The initial particle numbers within the isolates were subsequently determined using both the measured concentrations and the dilution factor. The particle-to-protein ratio was used to quantify the sEV isolate’s purity. The TEI-derived isolates had the highest protein contamination compared to UC and UCT (Figure 1D), indicating that the TEI isolates had the lowest purity, as demonstrated in Figure 1E. The UCT isolates exhibited the highest purity compared to the UC and TEI isolates (Figure 1E). Therefore, our findings suggest that a combination of half-cycles of UC and TEI may reduce the contamination ratio compared to TEI alone, while increasing the number of isolated particles compared to UC alone.

### 2.2. Comparison of Purity and Morphology of BC Cell-Derived sEVs and Common sEV Marker Expression

To investigate the expression of the sEVs obtained through the TEI, UC, and UCT methods, sEVs were captured in an equal number of particles, as determined by NTA using mixed CD9 and CD81 beads. The captured sEVs were then assessed using flow cytometry, employing a hybridised fluorescence detection of common sEV markers CD9, CD63, and CD81. Notably, the signal intensities of CD9, CD63, and CD81 in the UCT isolates were higher compared to the signal intensities of those in the other isolates, suggesting that UCT yielded the highest amount of sEVs in an equivalent number of particles (Figure 1F). Conversely, the signal intensities were similar between the UC and TEI isolates. To evaluate the sEV purity, the EV marker signal was analysed relative to the total protein ratio (Figure 1G). The graph clearly demonstrated that both the UC and UCT isolates exhibited higher purity than the TEI isolates, with the ratios of UC and UCT being significantly higher than that of TEI. Moreover, in the western blotting results, the levels of CD9, CD63, and CD81 were significantly higher in the UC isolates (Figure 1H). In contrast to the ratio presented in Figure 1G, the sEV marker expression in the UCT isolates appeared relatively lower compared to that in the UC isolates. The TEI isolates did not exhibit any detectable sEV markers, aligning with the findings in Figure 1G. This finding suggests that owing to the higher protein contamination levels, the amount of sEV proteins present in an equivalent protein amount could be lower in the TEI isolates compared to other isolates. Furthermore, none of the three methods exhibited an expression of Calnexin, with only the cell lysate (CL) showing a positive Calnexin expression.

The presence of sEVs in each isolate was verified through TEM analysis, which also included a morphological structure examination (Figure 2). The size of the sEVs varied considerably among the isolation methods and even within the isolates obtained through the same method. In the TEI isolates, cup-shaped sEVs (arrows), with an average size of around 100 nm, were observed alongside small non-EV vesicles (circles, <30 nm). These small vesicles could potentially be classified as lipoproteins. The UCT isolates also exhibited the presence of small vesicles, while the UC isolates predominantly consisted of cup-shaped sEVs without other small vesicles. In the UC isolates, the cup-shaped EVs ranged in size from approximately 100 nm to 200 nm. The UCT isolates displayed a wide range of cup-shaped EVs, varying from approximately 50 nm to 200 nm, indicating higher heterogeneity compared to the EVs isolated using the other two methods. It is assumed that UCT may be more suitable for isolating sEVs, as the other methods could result in the loss of the much smaller sEVs. Some sEVs in the UC isolates exhibited a damaged spherical shape, possibly due to the higher speed employed during the UC isolation process, leading to the aggregation of the EVs. Overall, based on the diverse characterisation of the sEV isolates, both the UC and UCT methods demonstrated higher-purity sEVs for the BC cell line compared to TEI. The sEV concentration was highest in the UCT isolates, while the expression intensity of common sEV markers was the strongest in the UC isolates.

### 2.3. Quantitative and Statistical Analysis of BC Cell-Derived sEV Proteins Identified by Three Different Isolations

To determine the optimal isolation method for sEV protein research, the label-free quantification was analysed for the total proteome in sEVs isolated using TEI, UC, and UCT. PCA (Principal Component analysis) score plot data were used to visualise consistent spectral patterns in each MD-MBA-231 cell-medium supernatant (Figure 3A). The three batches from each isolation method formed distinct clusters. The normalised abundance of the 2957 identified proteins was used for the score plot. PC1 explained 85.6% of the variance, while PC2 accounted for 9.2%. The 95% confidence ellipses revealed a tight cluster for TEI and UCT, whereas the UC cluster showed more variability between the batches. The cluster formed by TEI and UCT isolates was clearly distinguishable from the UC cluster.

To identify, based on MS data, the number of sEV-related proteins in the isolates from the different isolation methods, a Venn diagram was constructed. The UC isolates obtained the highest number of sEV proteins, followed by the UCT isolates (Figure 3B). Specifically, the UC, UCT, and TEI isolates obtained a total of 3105, 1021, and 964 proteins, respectively. Notably, the UC isolates had 1829 unique proteins compared to the UCT and TEI isolates, while the UCT and TEI isolates had only four and seven unique proteins, respectively. To identify the sEV proteins among the total proteins, Vesiclepedia and Exocarta databases were used. According to these databases, the UC isolates obtained 1966 sEV-related proteins, the largest number among the three methods. The UCT isolates contained 732 sEV proteins, while the TEI isolates showed 690 sEV proteins (Figure 3C,D). The UC isolates had 1066 distinct sEV proteins compared to the UCT and TEI isolates, while the UCT and TEI isolates had only one unique sEV protein each. Furthermore, there were 518 sEV proteins that overlapped among all three isolation methods.

### 2.4. Cellular Component of BC Cell-Derived sEV Proteins Identified Using GO Analysis

The sEV isolate proteins identified from the supernatant of the MDA-MB-231 cell medium were annotated using GO analysis using DAVID. Appendix A presents the top 10 most abundant GO terms associated with the isolates obtained from all three methods. The GO analysis revealed that the cellular components were largely similar across the three methods. The proteins obtained from the UC methods displayed the highest number of sEV-related genes compared to the proteins obtained through the other methods. The UC method also yielded a larger number of genes across all the cellular component terms owing to the extensive protein identification achieved through MS analysis. However, when considering the proportion of gene annotations within each cellular component, a higher proportion of isolated proteins in the UCT and TEI methods were annotated as sEV-related, accounting for 50.7% and 50.5% respectively, whereas the UC method only exhibited 38.5% of the proteins annotated as sEVs (Appendix A). Furthermore, the percentage of proteins related to the extracellular region and extracellular space was lower in the UC method compared to the UCT and TEI methods. Based on the GO analysis, it can be concluded that among the total identified proteins, the UC method has the lowest percentage of proteins related to sEVs, while the UCT method exhibits the highest percentage compared to the other methods. On the other hand, the GO analysis identified lipoprotein-related items, including chylomicron, very-low-density lipoprotein, low-density lipoprotein, intermediate-density lipoprotein, spherical high-density lipoprotein, and high-density lipoprotein particles. The TEI and UCT isolates contained a higher proportion of lipoprotein-related proteins compared to the UC isolates, as shown in Appendix A. Interestingly, only the UC isolates contained intermediate-density lipoproteins.

Overall, in terms of analysing the particle number in the MDA-MD-231 cell-derived sEV isolates, the UCT method demonstrated the highest purity and the presence of smaller particles. However, when specifically examining sEVs, the UC method exhibited higher sEV marker expression and purity. Moving onto the analysis of the entire proteome within the isolates, consistent patterns were observed across the replicates for the TEI and UCT isolates. In contrast, the UC isolates displayed variable patterns in terms of identified proteins across the different batches. When comparing the sEV proteins, the UC isolates showed a significantly higher number of sEV proteins compared to the other proteins, with minimal lipoprotein contamination.

These findings emphasise the importance of employing diverse characterisation methods to assess the quality and quantity of sEVs as well as the significance of selecting an appropriate isolation method, considering the advantages and disadvantages of each method, based on the downstream analysis requirements. Given that the aim of our study is to identify potential sEV protein biomarkers in BC, we have chosen to use the UC isolation method for the following experiments with cell lines.

### 2.5. Potential sEV Biomarkers Identified in MDA-MB-231, MCF-7, and SK-BR-3 BC Cell Lines

The UC method was used to isolate sEVs from three BC cell lines, namely MDA-MB-231, MCF-7, and SK-BR-3, as well as a normal breast cell line, MCF-10A, to identify potential sEV biomarkers by LC-MS/MS proteomics. Through the analysis, we identified 1463 significantly upregulated and 118 downregulated proteins in the MDA-MB-231 cell-line-derived sEVs, followed by 711 upregulated and 122 downregulated proteins in the MCF-7 cell-line-derived sEVs, and 295 upregulated and 192 downregulated proteins in the SK-BR-3 cell-line-derived sEVs (Figure 4A–C, respectively). A set of 218 commonly upregulated proteins and 32 commonly downregulated proteins was identified across all three BC cell lines (Figure 4D,E). These proteins were then matched against the Vesiclepedia and Exocarta databases, and 152 upregulated and 16 downregulated overlapped proteins were identified as specifically associated with sEVs (Figure 4F). To gain insight into the biological roles of distinctively expressed sEV proteins in BC cell lines, STRING analysis was applied, which revealed intercellular interactions and gene ontology categories (FDR = 0.01). The analysis uncovered functions, such as multivesicular body assembly and transport, which are likely involved in EV function and formation (Appendix A).

Additionally, we observed the involvement of the Notch signalling pathway, a known cancer-related function, as well as the complement and coagulation cascades, which are associated with the early immune response. It is important to note that while 158 sEV proteins exhibited significantly different expression levels in BC cell lines, some proteins did not appear in the STRING analysis owing to the absence of protein–protein interactions or significant annotations. In addition, 10 sEV proteins were identified, showing the most significantly different expression in the BC cell lines compared to the normal breast cell line (Appendix A). Downregulated sEV biomarker candidates include BDH2, INS, LAMA3, and TPX2, while upregulated ones include IFITM1, CEBPZ, CLTA, CHMP1A, VTA1, and SEC13. Upon validation, these proteins hold promise as potential sEV biomarkers in BC cell lines for BC diagnosis.

### 2.6. Characterisation of Human Plasma-Derived sEVs by Three Different Isolation Methods

The size distributions of the sEV particles derived from 300 µL of three BC patients’ plasma and three non-cancer individuals’ control plasma were analysed using NTA. Consistent with the findings from the cell-line experiments, the size distribution of the TEI isolates exhibited greater variability and a broader range compared to those of the other isolates (Figure 5A). The TEI isolates had a size range from approximately 35 to 500 nm, the UC isolates ranged from 95 to 325 nm, and the UCT isolates ranged from 75 to 325 nm. However, owing to the inherent complexity of plasma samples, the size distribution graphs displayed higher variability compared to those obtained from the cell lines. The UC and UCT isolates from human plasma appeared to exhibit greater consistency in the size distribution compared to that of the TEI isolates. The analysis of the mean mode size of the particles did not reveal any significant differences among the isolation methods (Figure 5B). However, the particle concentration was found to be significantly higher in the TEI isolates compared to the UC and UCT isolates (Figure 5C). Furthermore, the total protein amount (in µg/mL) was significantly lower in the UC isolates compared to the other isolates (Figure 5D), indicating that the UC isolates had the lowest contamination. The TEI isolates demonstrated the highest total protein concentration. Interestingly, the ratio of the particle number to the protein concentration did not show significant differences among the methods (Figure 5E).

### 2.7. Purity of Human Plasma-Derived sEVs and Common sEV Marker Expression Assessed by Flow Cytometry and Western Blotting

The expressions of CD9, CD63, and CD81 in human plasma samples were detected using flow cytometry. The expression level in the UC isolates was higher compared to that in the TEI isolates. Although the expression level in the UCT isolates was slightly higher than that in the TEI isolates, the difference was not statistically significant (Figure 5F). The ratio of the EV marker expression to the total protein was found to be highest in the UC isolates, indicating that the UC sEV isolates exhibited the highest purity (Figure 5G). The western blotting results demonstrated that the expressions of the Flotillin-1, CD9, and CD81 markers were the highest in the UC sEV isolates (Figure 5H), which is consistent with the results shown in Figure 5F. However, the CD63 expression was the strongest in the TEI isolates. The expressions for HSP70, Syntenin, and TSG101 were notably weak and exhibited similar levels of positivity across all the isolation methods. The Calnexin expression was only positive in CL.

In the TEM images, it was evident that all the isolates obtained from the three different methods exhibited distinct cup-shaped sEVs ranging in size from 100 nm to 200 nm (Figure 6). Notably, the isolates obtained using the TEI method displayed larger sEVs, reaching approximately 200 nm, along with a considerable presence of contaminants, such as non-EV vesicles and protein aggregation (depicted in black). On the other hand, the images obtained for the UC method exhibited a relatively clear background with a lower occurrence of contaminants. Similarly, the UCT images also revealed the presence of vesicles and protein aggregation in the background, but the amount of background contamination was less than that in the TEI images. Interestingly, the pattern of background contamination observed in the TEM images corresponded to the trend observed in the protein contamination graph displayed in Figure 5D. In summary, when assessing the characterisation and comparison of the sEVs isolated using the three different methods for human plasma samples, it was found that the UC method yielded the highest purity and strongest sEV marker expression.

### 2.8. Quantitative and Statistical Analysis of BC Plasma-Derived sEV Proteins Identified by Three Different Isolation Methods

In this study, we aimed to identify the most suitable sEV isolation method using three BC patients’ plasma samples (*n* = 3) through proteomic analysis. Following protein isolation from each of the three different isolation methods, which included TEI, UC, and UCT, PCA was conducted on the identified proteins from each isolation method, which demonstrated that the proteins from each method clustered together (Figure 7A). In contrast to the PCA score plot data of the BC cell line (Figure 3A), a tighter cluster of spectra was observed for the UC isolation method compared to the other two methods. The TEI method exhibited the highest variance for all three biological replicates, followed by UCT. These results were totally opposite to those of the PCA analysis of the BC cell-line-derived sEV isolates. The PC1 and PC2 variances were 77.3% and 11.9%, respectively. With 95% confidence ellipses, the TEI cluster is distinguishable from the UC and UCT clusters. Notably, 188, 169, and 191 proteins were identified in the UC, TEI, and UCT isolates, respectively, using quantitative analysis with Proteome Discoverer 2.4 (Figure 7B). Among all the identified proteins, the sEV-related proteins were selected using Vesiclepedia and Exocarta (Figure 7C). According to the databases, 135 sEV proteins were identified in the UC isolates, followed by 134 and 133 sEV proteins in the TEI and UCT isolates, respectively. Notably, there was no major difference in the number of sEV proteins found in the BC plasma among the three isolation methods. However, in terms of the sEV protein composition, the TEI isolates stand out with 13 distinct proteins, while the UC isolates exhibit eight unique proteins, and the UCT isolates show only two (Figure 7D).

The cellular components of the proteins identified using each isolation method were further analysed using GO terms (Appendix A). The GO analysis revealed that the nine most abundant cellular components were shared among the proteins isolated using all three methods. The sEV-related proteins, as the extracellular exosome term in UCT and UC, showed similar proportions, but the number of extracellular-region-related proteins was slightly higher in UCT, whilst TEI showed the highest percentage of extracellular-region-related proteins in the GO analysis. The lipoprotein-related proteins in the GO analysis demonstrated a similar pattern (Appendix A). The TEI isolates contained a higher percentage of lipoprotein-related proteins compared to the other two isolates, while the UC and UCT isolates had the same portion of lipoprotein-associated proteins. To assess the relative abundance of the sEV proteins, a comparison was conducted among those isolated via TEI, UC, and UCT. The analysis of the MS data revealed interesting findings, as presented in Appendix A, in the form of a heatmap. Remarkably, the large number of sEV proteins isolated using the TEI method displayed a significantly higher level of overexpression compared to the sEV proteins isolated using the other methods. Only a small part of the sEV proteins is highly expressed in UCT. These results indicate that all three isolation methods yielded distinct patterns of sEV protein expression, which may have implications for biomarker research. According to the GO analysis using DAVID, the highly expressed sEV proteins in the BC plasma-derived UCT isolates were related to the complement and coagulation cascade, a pathway associated with innate immune responses (FDR = 0.01) (Appendix A). On the other hand, the unique overexpressed sEV proteins in the UC isolates were linked to platelet function, immune response, phosphorylation, and focal adhesion (FDR = 0.01) (Appendix A). Although these findings may not serve as definitive guidelines for selecting an optimal isolation method, they can serve as a milestone for choosing an isolation approach based on the research area of interest.

### 2.9. Potential sEV Biomarkers and Their Biological Function in BC Plasma Samples

Owing to their similar percentages of lipoprotein-related proteins and sEV-related proteins, both UC and UCT were employed to identify potential sEV protein biomarkers. Biological replicates consisting of three BC patients and three non-cancer controls were utilised for this investigation. From the volcano plots, a significantly different expression was observed between the proteins isolated from the BC plasma and non-cancer control plasma using UC and UCT (*p* value = 0.05, fold change ± 1.5) (Figure 8A,B). Using UC, 12 upregulated and four downregulated plasma proteins were identified in the BC plasma samples. After further filtration based on the Exocarta and Vesiclepedia databases, 11 upregulated and three downregulated sEV proteins were found (Figure 8C and Appendix A). In the case of the UCT isolates, 23 upregulated and 18 downregulated proteins were highlighted in the BC plasma, for which seven were upregulated and 16 were downregulated sEV proteins (Figure 8D and Appendix A). Our STRING analysis revealed that the intercellular interaction clusters of sEV proteins isolated using UC were associated with the clotting cascade and lipoprotein particles, whereas those isolated using UCT were linked to the coagulation cascade and lipoprotein (Appendix A). It is important to note that certain proteins were not included in the STRING cluster and annotation owing to a lack of available protein–protein interaction data. Overall, our findings suggest that the UCT method retains a greater number of potential sEV protein biomarkers for BC in plasma compared to the other isolation techniques. The significantly overexpressed sEV proteins in the BC plasma-derived UCT isolates were ALB, COL1A1, CSN1S1, EEF1A2, and RPL3; and the C1S, C5, C7, and CFHR2 sEV proteins in the UCT isolates were the most downregulated proteins in the BC plasma compared to the non-cancer control. Those nine sEV proteins can be considered as potential sEV biomarkers for BC clinical diagnosis.

## 3. Discussion

sEVs are stable and abundant in various body fluids and are considered as a potential biomarker source for cancer diagnosis and monitoring cancer progression in real time. Currently, blood-derived sEV proteins hold promise as a prospective reservoir of novel biomarkers. Given that a blood sample is a complex mixture of various components and that blood-derived sEVs contain highly heterogenous information across patients, it is imperative to initially conduct a pilot study. There are a few ongoing clinical trials for sEV-based BC biomarker research. One study measures the expression of the HER2-HER3 dimer in plasma-derived sEVs to diagnose HER^+^ BC and guide the appropriate treatment (NCT04288141). Another clinical trial aims to validate the glycosylation of sEVs as a diagnostic biomarker for early BC diagnosis (NCT05417048). Two other studies focus on proteomic profiling within BC plasma-derived sEVs to identify a diagnostic biomarker for BC (NCT05798338) and a prognostic biomarker for neoadjuvant chemotherapy in BC treatment (NCT05831397). However, the lack of standardised methods for isolating sEVs has led to significant variability in both the quality and quantity of isolated sEVs, depending on the isolation methods and sample type. Consequently, the choice of the isolation method has a profound impact on the downstream analysis, leading to divergent outcomes. In this study, we provided a comprehensive qualitative and quantitative analysis comparing the isolated sEVs obtained through the TEI, UC, and UCT isolation methods. Our findings offer valuable insights and guidance for selecting the most suitable isolation method, particularly for proteomic analysis purposes.

In the conditioned medium of MBA-MB-231 cells, the UCT isolates demonstrated the most consistent particle isolation within the sEV size range. Consequently, the mode size of the particles obtained through UCT isolation was smaller than those of the particles obtained through the other methods, and the particle concentration was improved compared to the concentration of particles obtained from UC isolation alone. In terms of the particle number and protein contamination ratio, the UCT isolates exhibited higher particle numbers and similar protein contamination levels compared to the UC isolates, indicating the purest isolates in terms of the particle number and protein contamination ratio. However, it is important to note that these results do not directly indicate the purity and concentration of sEVs. The particle number per millilitre of the MDA-MB-231 cell-derived sEV isolates corresponded to the observations from the TEM images. The UCT isolates had a higher number of non-sEV vesicles compared to the UC isolates. In the flow cytometry analysis, the strongest expressions for CD9, CD63, and CD81 were detected in the UCT isolates, while western blotting showed that the set of sEV markers was the strongest in the UC isolates. During the flow cytometry process, the exclusive capture of sEVs by CD9- and CD81-conjugated beads could be the underlying reason for the potential loss of sEVs in the UC isolates that bear sEV surface markers different from those in CD9 and CD81. As a result, this loss of sEVs in UC isolates in the flow cytometry results may account for discrepancies in the overall expression of sEV markers observed in the western blotting results. It is important to emphasise that the UC isolates contained the highest number of sEV proteins compared to the other two isolates, even though the percentage of genes related to sEVs was the lowest. This is likely due to UC isolates containing the highest number of total proteins, which may also be related to the wider UC cluster observed in the principal component analysis. The proteins identified in the UC isolates exhibited more variability compared to those identified in the other isolates, resulting in a higher number of identified proteins.

No research has been conducted thus far comparing the proteomes of sEVs isolated through TEI, UC, and UCT across various BC cell lines. To the best of our knowledge, there is only one paper available in the literature that examined the difference between UC and TEI specifically in MDA-MB-231 cells, utilising western blotting to visualise the expression of sEV markers [26]. Interestingly, that study found that UC produced lower expression levels of CD9 and CD63 in MDA-MB-231 cells, which contradicts our results. However, the western blotting results in that study revealed variations in resolution, which could be attributed to differences in the exposure time, when assessing the expression of sEV markers across different isolation methods. This problem makes it challenging to directly compare the expressions between these methods. Furthermore, the study also revealed a greater abundance of sEV proteins in UC isolates compared to TEI isolates, aligning with the findings of our investigation. The variation in the specific number of identified proteins between the research and our investigation could potentially be attributed to differences in the MS spectra and the presence of contaminants within sEV isolates, which could be caused by different applications of enzymatic digestion and precipitation techniques used for isolating sEV proteins [24,27,28].

The overlapped potential sEV biomarkers across the BC cell lines were associated with several functions shown in the STRING network. Generally, cancer cells release a higher number of sEVs into the TME for intercellular communication, which transports various biological molecules, such as proteins and nucleic acids [29]. In the STRING network of the differentially expressed sEV proteins in the BC cell lines, some proteins were found to be related to amino acid transport across the plasma membrane and multivesicular body assembly, which corresponds to the increased number of sEVs in cancer and their function as intercellular communicators. Additionally, the retromer complex, which regulates endosome maturation and activates autophagy, was identified as another function [30]. Recently, it was discovered that the retromer complex might regulate the cargo in sEVs, thus influencing the neuronal function or disease [31], although its function in BC is unknown. Other biological roles of distinctly expressed sEV proteins include the complement and coagulation cascades, which are related to the innate immune response. The RHO GTPases are related to endosomal membrane trafficking in various cell lines [32] and sEV secretion in Oli-neu cells [33], which is responsible for the regulation of cancer invasion and progression [34]. Recently, it was discovered that the RHO GTPase family and its related lncRNAs, including RAD51-AS1 and DANCR, participated in BC carcinogenesis [35]. The Notch signalling pathway is one of the well-known factors in BC progression [36]. Moreover, sEV-encapsulated Notch ligands have been found in human umbilical vein endothelial cells and function to modulate Notch signalling by transferring the ligand to recipient cells [37]. In MCF-7-derived sEVs, a short form of Notch was discovered by proteomics analysis [38]. However, the role of protein cargoes carried by sEVs in Notch signalling in BC remain unclear.

In BC cell lines, a total of 152 sEV proteins were found to be upregulated, while 16 sEV proteins were downregulated. Upon analysis, we successfully identified the top 10 sEV proteins that displayed the most significant differences in their expression levels. The significantly downregulated sEV proteins in BC cell lines were BDH2, insulin (INS), LAMA3, and TPX2. INS is a well-known factor related to high-risk BC patients who have obesity [39,40]. Tissue biopsies of BC patients have shown that BDH2 has a positive relation with obesity in BC [41]. Interestingly, in MDA-MB-231 cells, INS-resistant adipocyte-derived sEVs promoted EMT, which was mediated by BRD2 [42]. This may suggest that INS and BRD2 play an important role in BC patients with obesity. Corresponding to our investigation, another study observed the low expression of LAMA3 in 20 BC cell lines, which might be related to tumour metastasis [43], but there was no information for sEV LAMA3 in BC. In a large cohort study, the nuclear expression of TPX2 was observed to be correlated with the clinical stage, negative ER and PR status in BC [44]. In contrast, in our current study, TPX2 was found to be significantly downregulated in sEVs derived from BC cell lines. The reason for this downregulation in sEVs and the functional implications of sEV TPX2 require further investigation in the future. Among the significantly upregulated potential biomarkers, it was found that IFITM1 was overexpressed in inflammatory BC cells [45]. However, the function and expression pattern of BC-related sEV IFITM1 have not been reported yet. The VTA1 protein binds to EV biogenesis-related proteins, an endosomal sorting complex required for transport (ESCRT), to facilitate EV formation [46], suggesting that VTA1 might cause an increase in the number of sEVs in BC. There is no information available regarding sEV CEMPZ, CLTA, CHMP1A, and SEC13 in BC. For the clinical application of these potential sEV biomarkers for BC diagnosis, further validation in clinical samples is required, and it needs to be clarified that those biomarker candidates are specific to BC, not for the entire spectrum of cancer.

In the human plasma samples, the characterisation of sEV particles yielded diverse results compared to those of the BC cell lines, likely due to the complexity of human samples. Notably, the TEI and UCT isolates exhibited various peaks in the size distribution. The UCT isolate had the largest portion of total particles within the largest size range. Paradoxically, the TEI isolates, which contained the highest abundance of protein contamination, exhibited the smallest particle size in the highest peak. The protein contamination in the UCT isolates from the plasma samples was higher than that in the UC isolates, while the BC cell results showed similar amounts between UC and UCT. Owing to the variability in the values across the plasma samples, the ratio of the particle number to the protein amount did not show any significant differences among the isolation methods. In contrast to the isolates from the BC cells, the UC isolates from the human plasma demonstrated noticeably higher purity than those of the isolates obtained using the other methods. Western blotting revealed weak signals for all the sEV marker proteins in the plasma isolates. This could be attributed to the higher protein contamination observed in the plasma isolates, as shown in the TEM results and GO analysis, making it challenging to detect specific sEV marker proteins in only 20 µg of proteins.

The proteomic analysis in this study provides additional evidence of distinct proteomic profiles in sEVs, depending on the sample type and isolation methods [15,19,47]. The PCA plots demonstrated that the TEI isolates from the BC plasma exhibited a broader cluster compared to the other two isolates, while the UC isolates showed a much tighter cluster. These findings suggest that the sEV proteome isolated using the TEI method demonstrated lower consistency compared to those isolated using the other two methods. Like the PCA plots, the GO analysis of the total protein in the isolates demonstrated similar proportions of sEV-related proteins and lipoprotein-related proteins in UC and UCT. The total number of identified proteins and sEV proteins in the plasma samples was significantly lower than that of identified proteins and sEV proteins in the BC cell lines. This is likely due to the use of frozen plasma from the biobank. Muller et al. found that sEVs in frozen plasma can be damaged during the freezing process, leading to decreased sEV purity [48]. Furthermore, as observed in this research, string-like aggregates were found in the TEM images of the plasma-derived sEVs in this study. Therefore, the ideal option would be to use fresh blood samples, although the frozen samples did not appear to affect the biological activity of the sEVs. In this study, to ensure comparability between the isolation methods, it was crucial to use the same plasma sample for all the methods, which allowed for guaranteed consistency and a valid comparison of the isolation techniques. In the top 10 significant GO terms for the cellular components of identified sEV proteins, the TEI isolates exhibited a higher percentage of sEV-related proteins compared to the UC and UCT isolates, with both the UC and UCT isolates showing similar percentages in the plasma samples. This result can be attributed to the relatively lower purity achieved using the TEI method compared to the other two methods, potentially resulting in fewer losses of sEVs but a higher presence of lipoprotein-related proteins. Both the UC and UCT isolates exhibited similar percentages of lipoprotein-related proteins in the plasma samples. However, to ensure a precise comparison of the abundance of lipoproteins, it is imperative to supplement the analysis with further validation through techniques such as ELISA or western blotting.

In a previous study, there was no report for validating UCT methods for isolating sEVs from human plasma samples. Only one paper demonstrated that UC isolates in human serum detected using western blotting showed more intense sEV marker expression compared to that of UCT isolates [16]. The innovation for our study is that we compared the isolation efficiency in terms of size, morphology, concentration, and purity for proteomic analysis in BC plasma.

The STING network analysis demonstrated that sEV biomarker candidates isolated using UC from BC plasma were associated with the clotting cascade, while those isolated using UCT were associated with the complement and coagulation cascades. The complement and coagulation cascades are involved in innate immune responses, which are a more plausible function for BC biomarkers. Interestingly, the potential biomarker cluster identified in both UC and UCT isolates had functions related to lipoprotein particles. One study suggested that the direct interaction between lipoproteins and EVs is an important factor in transferring biological information and EV uptake pathways [49]. Moreover, the increased uptake of lipoproteins facilitates the proliferation of BC cells [50]. Therefore, lipoproteins should be considered as potential sEV-related proteins and even potential sEV protein biomarkers rather than mere contamination in sEV isolates. Nonetheless, the absence of standardised isolation methods makes it challenging to separate lipoprotein particles from sEVs [51]. Consequently, establishing a definitive link between the isolated lipoproteins and sEVs, as opposed to contaminants, may require substantial efforts.

In BC plasma, the expressions of the sEV ALB, COL1A1, CSN1S1, EEF1A2, and RPL3 were significantly upregulated, while the expressions of C1S, C5, C7, and CFHR2 were downregulated, and these dysregulated proteins are considered as biomarker candidates for BC diagnosis. Although ALB has normally been known as contamination within sEV isolates, like lipoprotein, in sEV research [52], sEV ALB has also previously been found in BC patients’ plasma [53]. The role of ALB in BC plasma-derived sEV is still unclear, but low expressions of ALB in serum were found to be associated with poor BC prognosis [54]. Despite the previous discovery of COL1A1 [55], EEF1A2 [56,57], and RPL3 [58] as metastasis-related proteins in BC and CSN1S1 [59], C5 [60], and C7 [61] as BC prognosis-related proteins, the function or biomarker potential of these proteins within sEVs remains unknown.

The sEV proteins that have been identified in BC cell lines and in human BC plasma hold promise as markers in BC diagnosis, including APOA4, C3, C7, CLU, F2, and SERPINC1. These proteins are related to innate immune responses and the regulation of the insulin-like growth factor (IGF) transport and uptake by IGF-binding proteins. In a previous study, sEVs derived from BC cells were found to stimulate IGF-1 to induce transcription factors related to EMT in neighbouring cells [11], suggesting that those sEV proteins might influence the EMT process in BC mediated by the IGF transport. However, functional studies to uncover its specific role in BC metastasis and further validation will be mandatory for the practical application of these biomarkers in BC diagnosis. APOA4 plays a crucial role in HDL and VLDL metabolism [62]. Given the similar densities of sEVs and HDLs, their differentiation relies solely on size distinctions [63]. VLDL, being significantly smaller than sEVs, poses a challenge in achieving complete separation [64,65]. The possible contribution of APOA4 from lipoproteins rather than sEVs [66] will need to be established in future studies.

Recently, many advanced isolation methods have been developed. One of the techniques is asymmetrical-flow field-flow fractionation (AF4) [67]. The combination of UC and AF4 demonstrated efficient removal of lipoproteins from human serum samples [68]. Furthermore, for human plasma, the combination of immunoaffinity chromatography and AF4 successfully isolated exomeres, which are a smaller extracellular particle than sEVs, without any lipoprotein contaminants [69,70]. Methods such as AF4 and immunoaffinity chromatography combined with AF4 require additional specialised equipment and antibodies. The application of UCT is an accessible approach in a research laboratory and offers a cost-effective approach to isolate sEVs. However, it should be noted that lipoprotein and other vesicle-like contaminants are still present. This study’s focus was limited to the isolation and proteomics of sEVs. Given the challenges associated with isolating individual EV subpopulations, future studies may incorporate methods for eliminating vesicle-like contamination, particularly when the differentiation of the sub-populations of EVs is required.

## 4. Materials and Methods

### 4.1. Cell Lines and Cell Culture

The MDA-MB-231 triple-negative breast cancer (TNBC) cell line (RRID: CVCL_0062, ATCC^®^CRL-1435TM) was maintained in IMDM medium (Gibco, 12440061, Waltham, MA, USA) supplemented with 10% (*v*/*v*) foetal bovine serum (FBS) and 1% (*v*/*v*) antibiotics (50 U/mL of penicillin and 50 µg/mL of streptomycin). The SK-BR-3 HER^+^ BC cell line (RRID: CVCL_0033, ATCC^®^HTB-81TM) was cultured in McCoy’s 5a medium (Gibco, 16600082) supplemented with 10% (*v*/*v*) FBS and 1% (*v*/*v*) antibiotics. The MCF-7 ER/PR^+^ BC cell line (RRID: CVCL_0031, ATCC^®^CRL-1740TM) was cultured in RPMI-1640 medium (Gibco, 11875093) supplemented with 10% (*v*/*v*) FBS and 1% (*v*/*v*) antibiotics. The MCF-10A human mammary epithelial cell line (ATCC^®^HTB-81TM) was cultured using MEGM™ BULLETKIT™ (Lonza, CC-3150, Basel, Switzerland). All the cell lines that were used were negative for mycoplasma testing and were authenticated within the last three years through Short Tandem Repeat profiling by employing the PowerPlexR 18D System (Promega, Madison, WI, USA).

All the cell lines were cultured in a cell incubator (Thermo Scientific, Waltham, MA, USA) with 5% CO_2_ at 37 °C. When the cultures reached 80% confluency, the cells were washed with DPBS and detached with 0.25% trypsin-EDTA (1 mL per 25 cm^2^ of surface area) at 37 °C for 2 min. Two volumes of complete growth media were added, and the suspension was then transferred to a 15 mL Falcon tube and centrifuged at 1000 rpm for 5 min at room temperature (RT). The cells were resuspended in complete growth media for passaging.

### 4.2. Preparation of Cell Culture Medium for Extracellular Vesicle Experiment

Exosome-depleted cell media were appropriate cell media mixed with 10% (*v*/*v*) exosome-depleted FBS (Gibco, A2720801) and 1% (*v*/*v*) antibiotics (50 U/mL of penicillin and 50 µg/mL of streptomycin).

The BC cell lines were cultured until they reached a confluency of 60–70%. The cells were gently washed with DPBS twice and incubated with the exosome-depleted medium for 48 h (H). The medium was collected after 48 h to isolate the sEVs. The supernatant was collected and pre-purified by centrifugation to remove any cell debris. The supernatant was first centrifuged at 300× *g* for 5 min. It was then subjected to two more rounds of centrifugation at 2000× *g* for 20 min at 4 °C and at 10,000× *g* for 20 min. Finally, the supernatant was filtered through a 0.22 μm filter (Merck, Rahway, NJ, USA).

### 4.3. Plasma Sample Acquisition and Pre-Purification

All the plasma samples were obtained from the Health Precincts Biobank, which is a part of UNSW Biospecimen Services. A total of six individual plasma samples were used in this study, including 3 mL plasma samples from BC patients (*n* = 3) and 2 mL plasma samples from non-cancer controls (*n* = 3). Human ethics approval was obtained from UNSW Australia Human Research Ethics Advisory Panels (2022/HC220456) for using human plasma samples to conduct the EV analysis. The plasma samples were first centrifuged at 2000× *g* for 20 min at 4 °C followed by centrifugation at 10,000× *g* for 20 min at 4 °C. The final supernatant was filtered through a 0.22 μm filter and stored at −80 °C for downstream experiments.

### 4.4. sEV Isolation

To isolate sEVs from both the cell culture media and plasma, the TEI (Invitrogen (Waltham, MA, USA), 4478359 for the cell culture media and 4484450 for the plasma), UC, and a combination of UC and TEI (UCT) were used according to the manufacturers’ instructions.

TEI: The pre-purified cell supernatant was mixed with a 0.5 volume of the reagent and incubated overnight (o/n) from 2 °C to 8 °C. Following incubation, the sample was centrifuged at 10,000× *g* for 1 H from 2 °C to 8 °C. The resulting pellet was collected and resuspended in filtered PBS. Similarly, for the pre-purified plasma, the sample was mixed with a 0.5 volume of filtered PBS and a 0.2 volume of the reagent and incubated at RT for 10 min. After incubation, the sample was centrifuged at 10,000× *g* for 5 min at RT, and the pellet was collected in filtered PBS.

UC: UC was performed using a Beckman Optima XPN-100 and an L-100 XP (Beckman Coulter, Brea, CA, USA) equipped with Type 70 Ti and SW 55 Ti rotors. The pre-purified cell supernatant was ultracentrifuged at 100,000× *g* for 70 min twice, and the pellet was collected in PBS. The pre-purified plasma was ultracentrifuged at 120,000× *g* for 2 h twice before the pellet was collected in filtered PBS.

UCT combination method: The pre-purified cell supernatant was ultracentrifuged once at 100,000× *g* for 70 min using a Beckman Optima XPN-100, and the pellet was resuspended in 500 μL of filtered PBS. The redissolved pellet was mixed with a 0.5 volume of the TEI reagent and incubated o/n from 2 °C to 8 °C. The sample was centrifuged at 10,000× *g* for 1 h from 2 °C to 8 °C. For the plasma, samples were ultracentrifuged once at 120,000× *g* for 2 h using a Beckman Optima L-100 XP. The pellet was redissolved in 200 μL of filtered PBS, mixed with a 0.2 volume of the TEI reagent, and incubated for 10 min at RT. The sEVs were obtained as a pellet after centrifugation at 10,000× *g* for 5 min. For the sEV characterisation, the pellet was resuspended in filtered PBS.

### 4.5. Protein Quantification

A Qubit Protein Assay (Invitrogen, Q33211) was used to quantify the protein concentration following the manufacturer’s instructions. In brief, 10 μL of the isolates were combined with 190 μL of the Qubit working solution to yield a total volume of 200 μL. The mixture was then incubated at RT for 15 min, and the resulting solution was read using a Qubit fluorometer.

To quantify the protein content in the sEV isolates, a Pierce BCA protein assay kit (Thermo Scientific, 23225) was employed following the manufacturer’s instructions. To create a standard curve spanning a range of 0–500 µg/mL, nine points of serial dilution with BSA were prepared. The sample was diluted 1:10 in distilled water to yield a total volume of 150 μL, which was then mixed with 150 μL of the BCA working solution. All the samples and standard points were in replicates and incubated at 37 °C for 2 h. The absorbance of each sample was measured using a Synergy HT microplate reader (Bio-Tek, Winooski, VT, USA) at a wavelength of 562 nm, and the obtained absorbance values were converted to micrograms per millilitre using the standard curve. The final concentration was multiplied by the dilution factor to obtain the actual protein content in the sEV isolates.

### 4.6. Western Blotting

Ten or twenty micrograms of protein were separated on a 4–12% NuPAGE Bis-Tris gel (Invitrogen, NP0335BOX) and blotted onto 0.45 μm polyvinylidene fluoride membranes (Millipore/Merck, Rahway, NJ, USA). The membranes were incubated with 5% BSA for 1 H at RT for blocking. After blocking, the membranes were incubated with primary antibodies at 4 °C o/n and washed three times with TBST. The washed membranes were incubated with horseradish peroxidase (HRP)-conjugated secondary antibodies for 1 H at RT and washed three times. The membranes were then imaged using an enhanced chemiluminescence (ECL) substrate (Thermo Scientific, 34580) and ImageQuant LAS 4000 system (GE Healthcare, Chicago, IL, USA). Anti-rabbit (Cell signalling, 7074, Danvers, MA, USA) and anti-mouse (Cell signalling, 7076) secondary antibodies were used. Calnexin (Abcam, ab133615, Cambridge, UK), CD63 (Abcam, ab134045), CD9 (Abcam, ab263019), CD81 (Abcam, ab79559), Flotillin-1 (Abcam, ab133497), HSP70 (Abcam, ab181606), Syntenin (Abcam, ab133267), and TSG101 (Abcam, ab125011) were purchased from Abcam. All the antibodies were diluted at a ratio of 1:2000 except for CD81, which was diluted at 1:500.

### 4.7. Nanoparticle Tracking Analysis (NTA)

NTA was performed using a NanoSight NS300 system (NanoSight Technology, Malvern, UK) equipped with a 532 nm wavelength green laser and NTA software (NTA version 3.3; Malvern Instruments, Malvern, UK). The isolated EV samples from the cell supernatant were diluted with freshly filtered PBS (0.22 μm) and loaded into the detection chamber by a 1 mL syringe. The same settings were applied to all the samples: camera level: 9, detection threshold: 7, capture: 60 s, number of captured images: 5, and temperature: 25 °C. The original particle concentrations were calculated based on the measured concentrations and the dilution factor.

### 4.8. Transmission Electron Microscopy (TEM)

TEM was performed using a JEOL 1400 with a magnification scale ranging from 100 to 200 nm and a voltage of 100 kV. A two-hundred mesh carbon-coated copper grid (Ted Pella, Redding, CA, USA) was made hydrophilic by a glow discharge. Seven microlitres (μL) of each sample were absorbed in a grid for 10 min at RT, followed by negative staining with filtered 2% aqueous uranyl acetate.

### 4.9. Flow Cytometry

Immunomagnetic bead capturing was applied for accurate EV quantification in flow cytometry. The same number of particles in the EV samples measured by NTA were incubated with the following antibody-conjugated magnetic beads: 10 μL of anti-CD9 magnetic beads (2.7 μm, 1.0 × 10^7^/mL, Thermo Fisher Scientific, 10620D) and anti-CD81 magnetic beads (2.7 μm, 1.0 × 10^7^/mL, Thermo Fisher Scientific, 10622D) in 200 μL of PBS with 2% BSA o/n at 4 °C. The EV-captured beads were washed twice with PBS and 2% BSA and separated during the washing step using a MagJET separation rack (Thermo Fisher Scientific). The EVs captured by the magnetic beads were stained with the following antibodies: a mixture of APC-conjugated anti-CD9 (Thermo Fisher Scientific, A15698); anti-CD81 (Biolegend, 349510, San Diego, CA, USA); and anti-CD63 (Thermo Fisher Scientific, A15712) for hybridised EV detection. The isotype control was APC-conjugated anti-mouse IgG1 (Thermo Fisher Scientific, MA518093) and was incubated at 4 °C in the dark for 30 min. The beads were washed twice, and 500 μL of PBS was added. The sample was analysed using BD LSR Fortessa X-20 flow cytometry (Becton Dickinson, Franklin Lakes, NJ, USA). The data were processed using FlowJo software (Version 10). We used the ratio between the geometric means of the fluorescence intensity (gMFI) from each marker and the IgG isotype control to quantify each marker level.

### 4.10. Protein Digestion

The digestion was performed with RIPA buffer (Thermo Scientific, 89900) containing a proteinase and phosphatase inhibitor cocktail (100×) (Thermo Scientific, 78440). The proteins were precipitated using acetone precipitation. The proteins were reduced with 20 mM dithiothreitol and 20 mM ammonium bicarbonate at 37 °C for 30 min and alkylated with 40 mM iodoacetamide solution for 15 min at RT. Trypsin (Promega, V5111) was added to a final 1:100 (*wt*/*wt*) enzyme-to-protein ratio for o/n digestion. The digested peptides were purified using Pierce detergent removal spin columns, 125 μL, 25 columns (Thermo Fisher Scientific, 87776) and C18 tips (Thermo Fisher Scientific, SP301) and extracted in 80% ACN and 0.1% TFA.

### 4.11. Label-Free Quantitative Proteomics 

The digested peptides were solubilised in 10 µL of 0.1% formic acid and analysed by LC-MSMS using a Q Exactive Plus mass spectrometer (Thermo Scientific). The samples were loaded onto a micro-C18 precolumn (300 μm × 5 mm, Dionex, Sunnyvale, CA, USA) with H_2_O:CH_3_CN at 10 μL/min and then switched to a Valco 10-port valve (Dionex) in line with a fritless nano-column (75 μm × 20 cm) containing reverse-phase C18 media (1.9 μm, 120 Å, Dr. Maisch HPLC GmbH, Ammerbuch, Germany). The samples were eluted using a 120 min linear gradient of H_2_O:CH_3_CN (98:2, 0.1% formic acid) to H_2_O:CH_3_CN (64:36, 0.1% formic acid) at a flow rate of 200 μL/min. All the MS/MS spectra were obtained in data-dependent acquisition (DDA) mode from *m*/*z* 350 to 1750 at a resolution of 70,000 at *m*/*z* 200 and an accumulation target value of 106 ions. Up to ten of the most abundant ions (AGC target set to 105; minimum AGC target set to 1.5 × 10^4^) with charge states of ≥+2 and ≤+6 were fragmented by high-energy collision dissociation.

### 4.12. Proteomic Analysis

For the total proteomic analysis of the raw data, the data were analysed using Proteome Discoverer 2.4.1.15 with ion-based label-free quantification and compared against a *Homo sapiens* database (UniProtSwissProt, downloaded on Uniport, May 2022). The peptides in the raw data file were merged and identified with SEQUEST HT. The fixed modification was cysteine carboamidomethylation, and the variable modification was the oxidation of methionine. The digestion enzyme was trypsin with two missed tryptic cleavages. A 10 ppm peptide mass tolerance and a 0.6 Da fragment ion mass tolerance were used. The minimum peptide and maximum peptide lengths were 6 and 144, respectively. The false discovery rate (FDR) was less than 1%, as calculated using the Percolator algorithm.

### 4.13. Statistical Analysis

One-way ANOVA with multiple comparisons was used for the statistical analysis. The data were plotted as means ± standard deviations (SDs) and generated by GraphPad Prism (version 9.5.1, San Diego, CA, USA). The *p* value was considered as statistically significant. The statistically significant *p* values were indicated as follows: * *p* < 0.05, ** *p* < 0.01, *** *p* < 0.001, and **** *p* < 0.0001.

The principal component analysis (PCA) was performed using the “statistical analysis” tool of Metaboanalyst [71]. For the Gene Ontology (GO) analysis of the enrichment, DAVID (https://david.ncifcrf.gov/ (accessed on 20 April 2023) was used. The sEV-related proteins were identified using Funrich 3.1.4 [72] with the Exocarta database (released on 29 July 2015) and Vesiclepedia (version 4.1), limited to *Homo sapiens*. Venn diagrams were drawn using Funrich 3.1.4 and molbiotools (https://molbiotools.com/ (accessed on 2 May 2023). The biological role and intercellular interaction of the upregulated and downregulated sEV proteins in the BC cell lines and plasma were analysed using STRING (version 11.5) [73] of Cytoscape (version 3.9.1) [74] with the MCODE plugin (version 2.0.2) [75]. The interaction score was ≥0.4. The degree cutoff and node score cutoff were 2 and 0.2, respectively.

## 5. Conclusions

The two most frequently used sEV isolation methods, UC and TEI, are highly impacted by limitations, such as lower purities and sEV concentrations. In addition, there is currently no standardised method for sEV isolation specifically for proteomic analysis. In our current study, we thoroughly evaluated the combined approach compared to the sole UC and TEI methods, taking into account both the quantity and quality of the isolated sEVs, with a focus on suitability for proteomic investigations. Our findings provide valuable guidance for selecting the suitable approach among three isolation methods depending on the specific research objectives, sample types, and downstream analysis requirements. Regarding the exploration of sEV biomarkers for BC in clinical settings, our study has uncovered a solution to these problems through the implementation of a combined approach, which involves a half-cycle of UC followed by the TEI kit. This combination not only enhances the concentration of sEVs and efficiency of the sEV isolation process compared to traditional techniques but also identifies increased numbers of potential sEV protein biomarkers in BC plasma samples. These significant findings collectively demonstrate that the integration of UC and TEI isolation techniques significantly enhances the proteomic analysis of sEV-derived proteins in the context of plasma research. Although there have been many advanced isolation techniques, UCT can be easily implemented without the need for specialised equipment. To conclude, our study provides compelling evidence that the combination of UC and TEI isolation techniques is a promising method for isolating sEVs from human plasma samples and studying their potential protein biomarkers in BC. Moreover, in contrast to previous studies that demonstrated UCT’s efficiency using a straightforward approach, this paper stands alone in its comprehensive exploration of UCT isolation efficiency across multiple criteria, including size, morphology, concentration, and purity, specifically for proteomic analysis in human samples. Future studies are warranted to further optimise this method and validate its effectiveness in a large set of clinical cohorts.

## Figures and Tables

**Figure 1 ijms-24-15462-f001:**
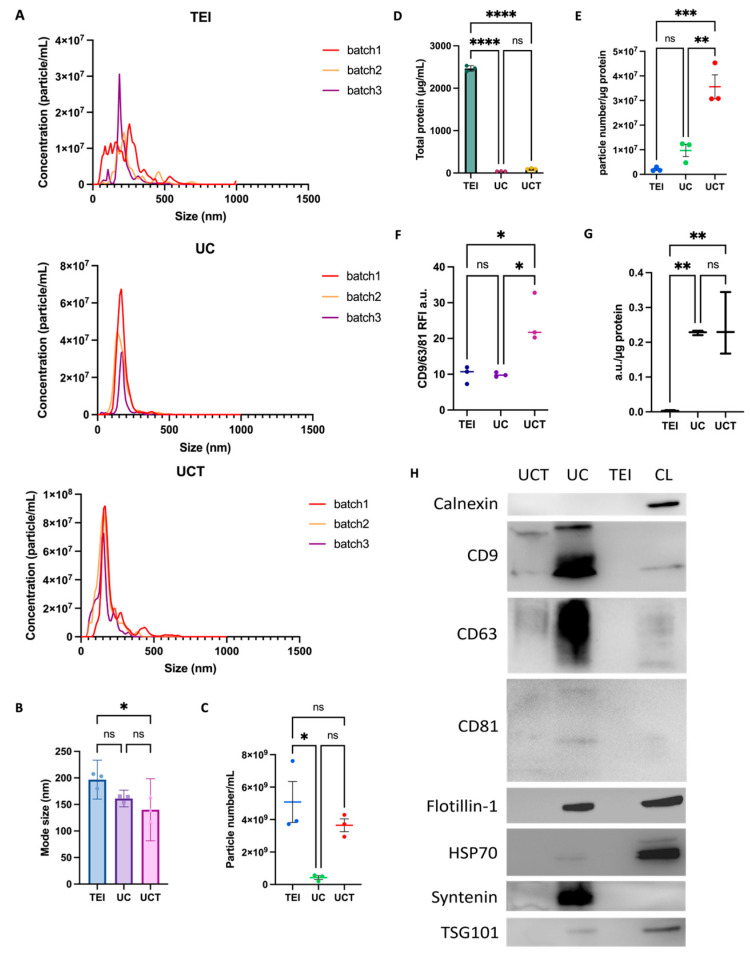
Characterisation of sEVs in MDA-MB-231 BC cell supernatant isolated using three different isolation methods. (**A**) The particle distribution, (**B**) mode size (nm), and (**C**) concentration (particle number/mL) of the isolates were analysed in three batches obtained through three different isolation methods using NTA. (**D**) Protein contamination (µg/mL) was measured in each isolate. (**E**) The particle number/protein ratio was calculated to assess the isolates’ purity. Each group consisted of three replicates (*n* = 3). (**F**) Flow cytometry data showing the presence of CD9, CD63, and CD81 expression with isolates in the same number of particles. Each dot presents each batch. (**G**) The ratios of the CD9, CD63, and CD81 signals to the protein concentration were used as an indicator of the sEV purity. (**H**) Typical expression of sEV markers (CD9, CD63, CD81, Flotillin-1, HSP70, Syntenin, and TSG101) and the negative protein (Calnexin) detected in western blotting. For each gel lane, 10 µg of cell lysate proteins were used as a positive control for Calnexin, and proteins isolated using UC, UCT, and TEI isolation methods were loaded. Each dot in (**B**,**C**,**E**,**F**) represents a single batch. The data are presented as mean ± standard deviation (SD), and statistical analysis was performed using one-way analysis of variance (ANOVA) with multiple comparisons. The significance levels (* *p* < 0.05, ** *p* < 0.01, *** *p* < 0.001, and **** *p* < 0.0001) indicate the level of statistical significance. Abbreviations: BC: breast cancer; CL: cell lysate; ns: non-significant; NTA: nanoparticle tracking analysis; sEVs: small extracellular vesicles; TEI: total exosome isolation kit; UC: ultracentrifugation; UCT: ultracentrifugation followed by total exosome isolation kit.

**Figure 2 ijms-24-15462-f002:**
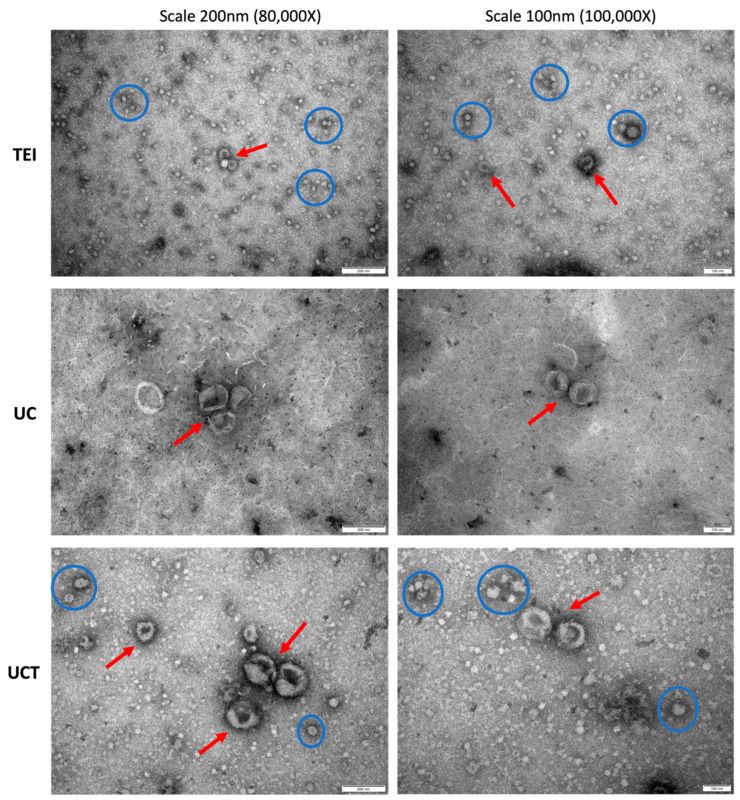
Representative TEM images of morphological characterisation of sEVs isolated from a BC cell line by three different isolation methods. TEM images with negative staining were captured to visualise the cup-shaped sEVs (arrows) and small particles potentially classified as lipoproteins (circles) in the MDA-MB-231 cell line. The isolates from the TEI, UC, and UCT methods were imaged at magnifications of ×80,000 and ×100,000, with scale bars of 200 nm and 100 nm used for the left and right columnar images, respectively. The representative images were selected from a minimum of 10 images for each isolate obtained from the different methods. The red arrows indicate sEVs, and the blue circles indicate non-EV vesicles. Abbreviations: sEVs: small extracellular vesicles; TEI: total exosome isolation kit; TEM: transmission electron microscopy; UC: ultracentrifugation; UCT: ultracentrifugation followed by total exosome isolation kit.

**Figure 3 ijms-24-15462-f003:**
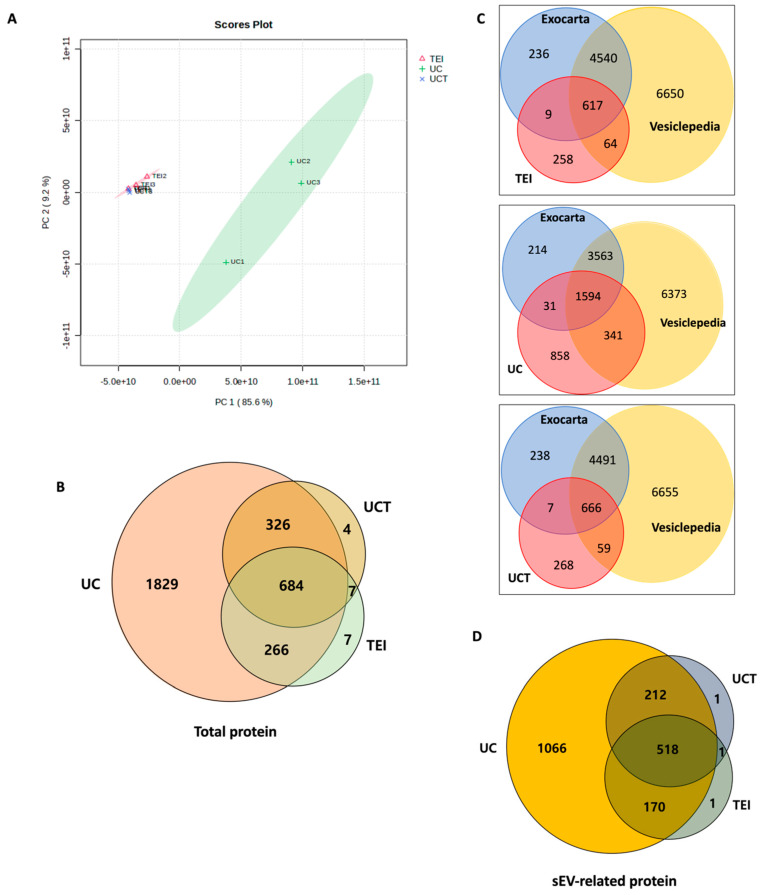
Proteome identified in MDA-MB-231 cell-line-derived sEV isolates by TEI, UC, and UCT isolation methods. (**A**) PCA score plot for proteins identified in each batch of the conditioned medium of the MDA-MB-231 BC cell line by mass spectrometry. The plot shows PC1 versus PC2 as a percentage with 95% confidence ellipses. (**B**) The total number of proteins in isolates from UC, TEI, and UCT based on the mass spectrometry data. (**C**) The Venn diagram showing the number of sEV-related proteins based on Vesiclepedia and Exocarta data. (**D**) Number of sEV-related proteins based on Exocarta and Vesiclepedia data for TEI, UC, and UCT isolates are compared in Venn diagram. Abbreviations: PCA: principal component analysis; sEV: small extracellular vesicle; TEI: total exosome isolation kit; UC: ultracentrifugation; UCT: ultracentrifugation followed by total exosome isolation kit.

**Figure 4 ijms-24-15462-f004:**
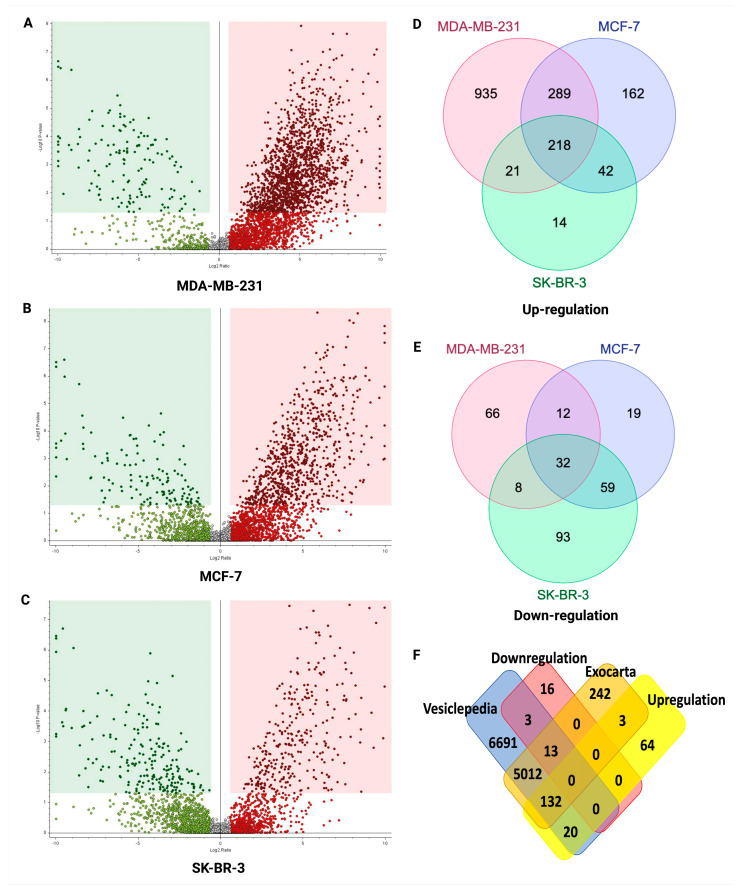
The volcano plots represent the quantitative analyses of the proteomes of sEVs in three BC cell lines vs. one normal breast cell line isolated using UC. Venn diagram showing potential sEV biomarkers for BC. (**A**–**C**) Significantly upregulated (red dot) and downregulated (green dot) sEV proteins in MDA-MB231, MCF-7, and SK-BR-3 cell lines identified via quantitative and statistical analysis via Proteome Discoverer 2.4 (*p* value = 0.05, fold change ± 1.5) based on three biological replicates (*n* = 3), respectively. (**D**,**E**) The 218 common upregulated proteins and 32 downregulated proteins were identified in three BC cell lines, respectively. (**F**) The Venn diagram represents 155 upregulated sEV proteins and 16 downregulated sEV proteins according to the Vesiclepedia and Exocarta databases. Abbreviations: BC: breast cancer; sEV: small extracellular vesicle.

**Figure 5 ijms-24-15462-f005:**
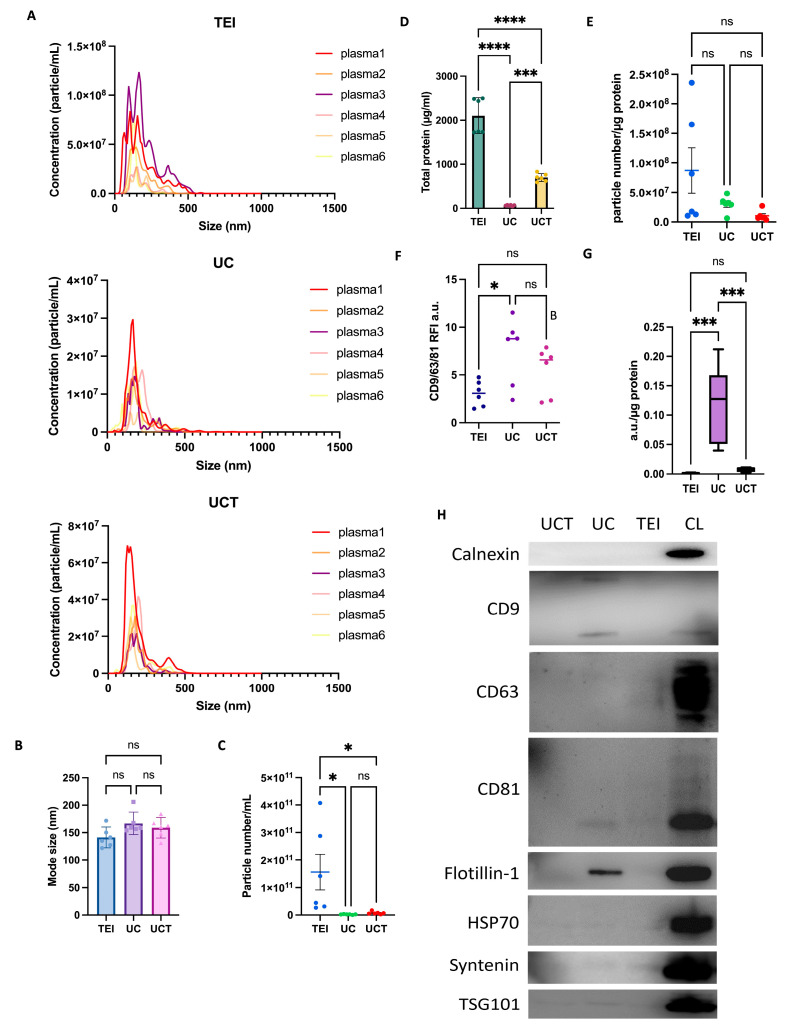
Characterisation of sEVs in human plasma samples isolated using three different isolation methods. (**A**) Particle size distribution, (**B**) mode size (nm), and (**C**) concentration (particle number/mL) of isolates in six plasma samples (Total number of samples = 6, BC plasma = 3, and non-cancer plasma = 3) isolated using TEI, UC, and UCT and nanoparticle tracking analysis. (**D**) Protein contamination (µg/mL). (**E**) Particle number/protein ratio for each plasma sample indicates sEV purity. (**F**) The expressions of CD9, CD63, and CD81 in isolates containing the same number of particles were detected using flow cytometry (*n* = 6, BC plasma = 3, and non-cancer plasma = 3). (**G**) The ratio of the CD9, CD63, and CD81 signals and protein concentration shows the sEV purity. Each dot represents a single plasma sample in (**B**,**C**,**E**,**F**). Data are shown as mean ± SD, and one-way ANOVA with multiple comparisons was used in (**B**–**G**). * *p* < 0.05, *** *p* < 0.001, and **** *p* < 0.0001. (**H**) The expressions of CD9, CD63, CD81, Flotillin-1, HSP70, Syntenin, and TSG101 as the common sEV markers and Calnexin as a negative control marker were detected using western blotting. A total of 20 µg of cell lysates and proteins of isolates obtained from UC, UCT, and TEI methods were loaded in each gel lane. Abbreviations: BC: breast cancer; ns: non-significant; sEV: small extracellular vesicle; TEI: total exosome isolation kit; UC: ultracentrifugation; UCT: ultracentrifugation followed by total exosome isolation kit.

**Figure 6 ijms-24-15462-f006:**
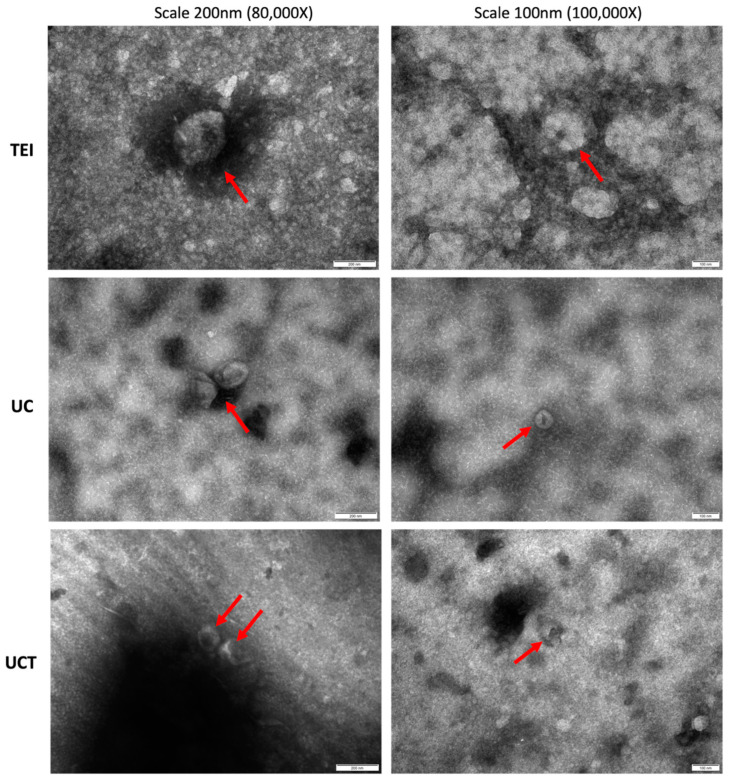
TEM images showing morphological characterisation of sEVs isolated using three different isolation methods. Negatively stained grid demonstrated the TEM images of cup-shaped sEVs (red arrows) in plasma samples isolated using TEI, UC, and UCT. Scale bars are 200 nm for the images in left column and 100 nm for right-column images. The magnifications are ×80,000 for top row images and ×100,000 for bottom row images. All the images are representative images out of at least 10 images of each isolate obtained from the three methods. Abbreviations: sEV: small extracellular vesicle; TEI: total exosome isolation kit; TEM: transmission electron microscopy; UC: ultracentrifugation; UCT: ultracentrifugation followed by total exosome isolation kit.

**Figure 7 ijms-24-15462-f007:**
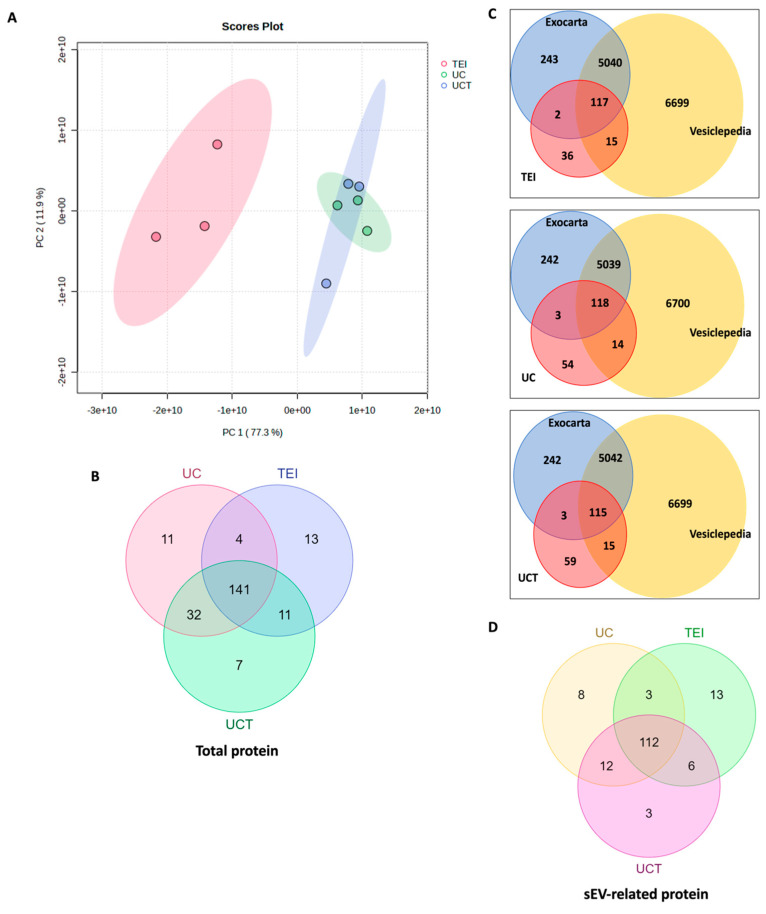
Proteomic analysis of proteins from BC patients’ plasma isolated using three different isolation methods and identified using mass spectrometry. (**A**) PCA score plot shows the variance in proteins identified in three BC patients’ plasma samples. The proteins were isolated using TEI, UC, and UCT isolation methods and identified using MS. The plot shows PC1 versus PC2 as a percentage, with 95% confidence ellipses. (**B**) The number of total BC plasma proteins identified from the UC, TEI, and UCT isolates is based on the MS data. (**C**) The Venn diagram represents numerous sEV-related proteins listed in Vesiclepedia and Exocarta databases. (**D**) sEV-related proteins identified using UCT, TEI, and UC isolation methods were compared in Venn diagram. Abbreviations: BC: breast cancer; MS: mass spectrometry; PCA: principal component analysis; sEV: small extracellular vesicle; TEI: total exosome isolation kit; UC: ultracentrifugation; UCT: ultracentrifugation followed by total exosome isolation kit.

**Figure 8 ijms-24-15462-f008:**
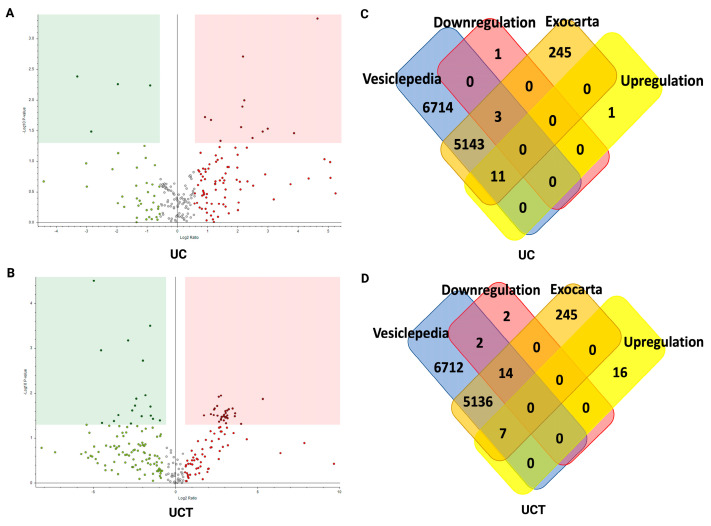
The volcano plots representing the quantitative analyses of the sEV proteomes isolated using UC and UCT in BC patients vs. non-cancer controls. Venn diagram showing potential of sEV biomarkers for BC screening. Significantly upregulated (red dot) and downregulated (green dot) proteins in BC patients’ plasma isolated using UC (**A**) and UCT (**B**) were identified via quantitative analysis using Proteome Discoverer 2.4 (*p* value = 0.05, fold change ± 1.5) for three biological replicates. (**C**) The Venn diagrams represent 11 upregulated and three downregulated sEV proteins isolated using the UC method. (**D**) Seven upregulated and 16 downregulated sEV proteins were isolated using UCT. Abbreviations: BC: breast cancer; sEV: small extracellular vesicle; UC: ultracentrifugation; UCT: ultracentrifugation followed by total exosome isolation kit.

## Data Availability

All the data are available on request.

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
