# Peer review of "Comparison Study of Small Extracellular Vesicle Isolation Methods for Profiling Protein Biomarkers in Breast Cancer Liquid Biopsies"

_ijms, 2023, doi:10.3390/ijms242015462_

Round 1
Reviewer 1 Report
An interesting manuscript. Some questions are raised for your consideration.
1. Page 4, Figure 1: The authors should elucidate the normalization process for determining the particle number from Nanoparticle Tracking Analysis (NTA) to establish the particle/total protein ratio.
2. Page 4, Figure 1H: Is any target detected for the TEI sample? The authors should include a loading control for the western blot to confirm equal protein loading.
3. Page 11, Figure 5H: Similar to comment 2, a loading control is required.
4. Page 19, lines 597-598: There appears to be a discrepancy between the conclusions drawn from the proteomic analysis and the observations in Figure 5. The author claimed that “TEI isolates exhibited a higher percentage of sEV-related proteins, while UC and UCT isolates showed similar percentages.” However, in Figure 5, UC isolates were identified to have the highest purity. Could the author elaborate on this subject?
Reviewer 2 Report
The authors have conducted a nice study where they compare isolating with UC, precipitation, and a combination of these techniques. The study is well constructed, however, there are some questions that the authors should address.
Lines 66-68: Authors make this claim about isolation, however, there are many potentially better techniques than UC and precipitation, and this should be addressed either in introduction or discussion section. See more information in https://doi.org/10.1016/j.chroma.2020.461773
Introduction: why UC and TEI have been chosen since they are well known for producing impure EVs and have other limitations especially if clinical applications would be of interest.
Lines 93-96: Why do authors present results in the introduction section?
The introduction or discussion section needs to be expanded to critique own choice of isolation methodologies and compare them to state-of-the-art in terms of purity from lipoprotein and other protein contaminants and, the possibility to specify the isolated subpopulations reproducibly.
Section 2.1 Has NTA been compared with other size distribution techniques, since NTA is known to not produce reliable results? https://www.oxfordglobal.co.uk/wp-content/uploads/2022/07/Nanoview-Technical-Note-NTA-Comparison.pdf
Line 117: The purity difference seems to be minimal since they are all in the same order of magnitude. Might be easier to just do the UC? Monolithic affinity chromatography can produce EV samples with particle:protein ration of 10^9 from plasma, which would be 100 times more enriched compared to UCT and have no contaminating proteins, and well-defined size distribution. Isolation time is also under 10 min per plasma sample.
Lines 155-159: What is the amount of lipoprotein contaminants?
Lines 168-170: This heterogeneity seems to be a negative thing? Is it reproducibly heterogeneous?
Line 165: You say could potentially be classified as lipoproteins, however in figure 2 you classify them as lipoproteins.
Lines 302-303. I would assume that UC and UCT also have particles under 75 and 95 nm, this is just issue with NTA that it doesn’t notice those. Comment on this? Did you see smaller particles in TEM for example? NTA is not specific to EVs, so if you see lipoproteins and other proteins in TEM, you might suspect that the NTA results are not giving correct values at lower end of size distribution.
Figure 5. You are missing text in 5E. The purity difference seems to be minimal since they are all in the same order of magnitude. Might be easier to just do the UC? In addition, monolithic affinity chromatography can produce EV samples with particle:protein ration of 10^9 from plasma, which would be 100 times more enriched compared to UCT and have no contaminating lipoproteins, and well-defined size distribution. Isolation time is also under 10 min per plasma sample.
Lines 588-590: Could this also be the result of the methodology. So that the methods selected are not good enough for plasma isolation of EVs?
Line 598-600: Lipoprotein identification is missing. For example, ELISA for Apob, apoa and apoe. This is crucial, since even your NTA results are not reliable since there are abundantly lipoproteins like LDL and VLDL that could distort the results.
Lines 603-605: Why would you recommend UCT as a method compared to other more efficient isolation systems? What are innovations of the method in the study compared to automated isolation system? https://doi.org/10.1021/acs.analchem.0c01986
Lines 614-616: Then you need to show the difference between isolated lipoproteins, lipoprotein free EVs, and EVs isolates “contaminated” with lipoproteins to make this kind of claim. Lipoproteins are huge nanosized particles, not small proteins, so for them to be sEV-related they need to be internalized or attached to the surface of EVs. This needs to be shown experimentally, to make sure that it is not an artifact of UC centrifugal forces or some other methodological artifact. For example, confirmed with many different methods.
Lines 627-628: isn’t APOA4 an apolipoprotein found on lipoproteins such as HDL and VLDL. Why would we use this as biomarker for BC diagnosis and isolate sEVs for this biomarker, if we could just isolate HDL and VLDL, which is much easier to do.
Lines 816-818: How does this study help to select the most suitable approach if there is only UC and precipitation method studied? There might be much better techniques available.
Lines 820-823: you talk about conventional techniques; however, you are in fact using conventional techniques, just slightly modified.
Lines 826-830: Is this the only study that combines UC and TEI? if not elaborate how is this better than previous efforts, if disregarding the BC.
Round 2
Reviewer 1 Report
I am pleased to report that, after evaluation, it appears that the authors have addressed all the comments and questions that were previously raised. The revision have significantly improved the clarity and quality of the manuscript.